# The Dynamic Evaluation Model of Health Sustainability under MCDM Benchmarking Health Indicator Standards

**DOI:** 10.3390/ijerph20010259

**Published:** 2022-12-24

**Authors:** Nutthawut Ritmak, Wanchai Rattanawong, Varin Vongmanee

**Affiliations:** 1Graduate School, University of the Thai Chamber of Commerce, 126/1 Vibhavadi Rangsit Road, Din Daeng, Bangkok 10400, Thailand; 2School of Engineering, University of the Thai Chamber of Commerce, Bangkok 10400, Thailand

**Keywords:** health sustainability model, evaluation, MCDM, AHP, TOPSIS

## Abstract

The coronavirus pandemic causing millions of deaths around the world has raised awareness of the importance of healthcare in a city, especially its efficiency in the city’s system. Although the health dimension is found to have critical effects on sustainable development in addition to the existing three dimensions, the majority of sustainability assessments that are developed based on the UN’s Sustainable Development Goals (SDGs) do not define standards for each indicator. On the contrary, an effective assessment model should be dynamic and suitable to the context of each city by referring to standardized criteria, such as those in a health indicator standard (HIS), instead of comparing with other cities. Hence, this research presents a new sustainability assessment model that integrates the health dimension (HEDm) with the other three dimensions generally used in studies of sustainable development (STD) and develops an assessment method to evaluate real data with references to the HIS from reliable organizations, such as the World Health Organization (WHO) and the United Nations (UN), for the SDGs. By using the Technique for Order of Preference by Similarity to Ideal Solution (TOPSIS) with the framework obtained from literature reviews, ISO Standards (ISO37120, ISO37122, ISO37123) and U4SSC, the researchers formed indicators in four dimensions with a total of 15 elements and 45 indicators. Afterwards, the weight of each indicator was determined with the Analytic Hierarchy Process (AHP) conducted in each dimension and element, resulting in a four-dimensional sustainability assessment model based on HIS value that limits the need to compare with other cities. The results from Khon Kaen province have shown the weaknesses and strengths of the target city that policymakers can reference to formulate a policy and strategy to improve HIS values in each dimension to meet standards, elevate city capacity and raise quality of life for the people living in the area.

## 1. Introduction

STD was first presented in World Conservation Strategy, which was jointly published by the International Union for Conservation of Nature (IUCN), the United Nations Environment Program (UNEP) and the World Wildlife Fund (WWF) in 1980 [1]. As given in the text, “Sustainable development is development that meets the needs of the prevention without compromising the ability of future generations to meet their own needs” [2]. Later, in 2000, the United Nations (UN) proposed the eight Millennium Development Goals (MDGs) to develop quality of life (QoL): 1. to eliminate extreme poverty and hunger; 2. to achieve universal primary education; 3. to promote gender equality and empower women; 4. to reduce child mortality; 5. to promote maternal health; 6. to fight malaria, HIV/AIDS and other diseases; 7. to promote environmental sustainability; 8. to develop a universal partnership for development. This was the official beginning of the integration of health and sustainability universally [3]. However, the MDGs were ended in 2015; in their place, the UN established “the Sustainable Development Goals (SDGs)” as a new target framework to drive the global STD until August 2030 (a period of 15 years). The SDGs were developed based on three dimensions of sustainability: the economic dimension (ECDm), social dimension (SODm) and environment dimension (ENDm), which all encompass 17 goals [4]. From the SDGs, the researchers have integrated STD into various dimensions of social studies such as education, business, tourism, politics, culture and locality development. Sustainability is an ideal state when setting goals towards STD, and it should be reflected in an international policy and strategy for development to fulfill basic human needs, stimulate economic growth and promote environmental preservation. STD is adjustable, depending on the goals and contexts of cities. In general, sustainability indices measure efficiency in sustainability assessment and provide information on the effectiveness, weaknesses and strengths of a city’s administration. The data enable a city to direct budget for STD in accordance with the problems and requirements at the time.

Based on reviews of the literature as shown in Table 1, researchers have found that although a number of studies on sustainability assessment have been conducted at the city level, regional level, national level and international level, there still has not been an area-specific study targeted towards localities with unique characteristics [2]. Moreover, those studies regularly aim at answering which cities are the most sustainable by making a comparison with other cities under entirely different contexts and characteristics. As a result, the comparisons lead to skepticism regarding whether the most sustainable city is truly successful, leaving room for future research to attentively improve assessing mechanisms. Some studies have proposed the concept of sustainability assessment for a city with multi-criteria decision making (MCDM), using a TOPSIS analysis developed from adjustable indices to evaluate the foundations and targets of a city and explore its specific answers [5]. The concept results in higher efficiency from actual assessment, allowing the results to be utilized and form local STD goals from a bottom-up approach based on the problem identification method. Considering present conditions, more than 50% of the world’s population lives in cities [6], and the number will increase to 70% by 2050 [7]. Cities are facing challenges in multiple aspects such as inadequate infrastructure, economic inequality, insufficient resources and effects from environmental abuse [8]. Smart evaluation and design of a city, therefore, leads to sustainability in health, the environment and the economy in the city [9,10,11,12]. Table 1 shows that the index set by international organizations aims at developing a theory for sustainable development. They are widely used in assessments of the dimensions of economy, society and environment; however, those assessments do not respond to the intrinsic needs of each country, especially emerging countries or developing countries that heavily focus on economic development [13]. These assessments also neglect social and environmental sustainability that, in fact, cause an impact to quality of life in a city [14]. In addition, academics frequently use qualitative methodologies to study city development processes and quantitative methodologies to study the sustainability of a city [15]. To compare value among multiple cities, MCDM is frequently used due to its suitability in a multi-dimensional assessment with a quantitative approach; however, the method lacks dynamic assessment that is modifiable based on criteria and standards at the time [16]. Those studies highlighted an overall development of sustainability, whereas only limited numbers of researchers conducted multi-level assessments [2]. Moreover, as the model used in the assessment provides only one option to demonstrate the most sustainable city, it cannot offer in-depth knowledge on sustainability by dimension and element. This is in contrast with the fact that an appropriate model should be able to present data in multiple forms by demonstrating data in various dimensions and elements [16]. Furthermore, the existing model widely assigns weights obtained from experts’ opinions that may contain some biases [17]. While most research primarily focuses on assessments from three dimensions of sustainability, the pandemic that recently occurred has clearly illustrated that health substantially affects all dimensions of sustainability. Therefore, sustainability assessment after the pandemic era should integrate the third goal of the SDGs, Good Health and Well Being, to ensure healthy livelihoods and promote wellbeing for all at all ages by covering issues on health and safety while complying with WHO frameworks.

To fill all of the gaps, research was conducted to present dynamic assessments from four dimensions, health, economy, society and environment, for a city with reference to the MCDM Benchmarking Health Indicator Standard. Though the task to determine the positive ideal and negative ideal is challenging and time-consuming, as it requires the researchers to gather HIS values from reliable organizations for all 45 indicators, it has allowed the researchers to identify indicators that needed to be improved for the target city and had not been clearly shown in previous studies. Moreover, the model shown in this research is highly accurate without any overestimation and functional even in the case with one alternative while other research that uses other MDCMs, such as AHP-TOPSIS requires more than one alternative. In addition, while other research has not been able to identify whether each IDC meets the desired HIS value, this study proposes a city assessment model to identify the points that must be improved in dimensions, elements and IDCs to develop a city up to the preset standards. It also enables policymakers to understand problems from both an overall view and in detail, leading to effective policy formulation in city development. The paper aims to present dynamic evaluation model of health sustainability under MCDM with reference to health indicator standards to quantitatively assess sustainability from the health dimension together with the other three dimensions. The study consists of five sections. Section 1 introduces literature reviews. Section 2 elaborates on the methodologies of the study, starting from the selection of IDC, compilation of HIS and weight determination of each IDC and ending with the TOPSIS analysis. Then, Section 3 shows the results of the target city, Khon Kaen province in Thailand, and Section 4 and Section 5 conclude the study with a discussion of limitations for future research.

## 2. Materials and Methods

The materials and methods for the conceptual framework of this research is summarized in Figure 1.

### 2.1. Establishing Criteria and Standard

The objective of this research is not to rank which city is the most sustainable, but to focus on sustainability assessment in four dimensions: health, economy, society and environment, together with elements to gain in-depth understanding of the strengths and weaknesses of a city.

#### 2.1.1. Selection of Indicators in Forming a Model

The researchers gather indicators from literature reviews and standards in relation to building a sustainable city, such as ISO37120, ISO37122, ISO37123 and U4SSC [27,28,29,30,31] to select criteria (CTR), hereafter called indicators (IDC) with consideration for the context of Thailand. The selected indicators must consist of quantitative secondary data disclosed by the public sector because TOPSIS analysis requires quantitative data.To verify suitability with the index of item objective congruence (IOC) by closed-ended questions on the indicators derived from reviews of the literature, an odd number of experts were invited. For each indicator, each expert was asked for its suitability. The possible answers were minus one (−1) if the expert deemed it unsuitable, zero (0) if the expert was uncertain or plus one (+1) if the expert deemed it suitable. The value for each indicator must be close to the minimum value of 0.05, whereas the indicators with a value less than 0.05 must be improved or deleted [32]. The CVs of the experts are shown in Table A1.

#### 2.1.2. Data Compilation for TOPSIS Analysis

The process started by assigning a weight ($w_{j}$) to each indicator of the TOPSIS analysis. This was determined by asking the same group of 15 experts their opinion on the rate of suitability of each indicator using an AHP pairwise comparison matrix based on Saaty’s nine point ratio scale. This was done anonymously through an online conference to mitigate issues of personal and group bias and maintain independence of opinions.The researchers compiled the data as of 2019 for the target city from both online and offline government databases to standardize the base year and maintain consistency with data from IDCs. All data and their sources are shown in Table A2.The researchers compiled health indicator standards (HIS) set by reliable international organizations, public policies and research with a credible database. The data used for HISes are also as of 2019 to maintain neutrality, as it is the year before the occurrence of the pandemic, and to ensure the completeness of the data.

### 2.2. Analytic Hierarchy Process (AHP)

AHP, developed by Thomas L. Saaty [33], was used in analyzing IDCs. By using AHP, each IDC was analyzed quantitatively and qualitatively with rational psychology to attributing values. Each value, which is independent from the opinions of experts, was tested with a consistency test and shown as a consistency ratio (*C.R.*) to validate an eigenvector (*EV*). Each value was deemed acceptable when $w_{j}$ was equal to or less than one after the following calculation steps.

#### 2.2.1. Pairwise Comparison Matrix (PCM)

First, the researchers create a PCM with the size of n×n where n is the number of IDCs to identify the $w_{j}$ of each IDC pair. Let $c_{i}$ be the main decision-making IDC, where $A_{j}$ is the IDC in the secondary hierarchy to analyze $a_{ij}$. Then, compare all $c_{j}$ and $A_{i}$ pairs with Saaty’s nine point ratio scale until completion, with the results described in an *i* × *j* matrix where i and j = 1,2,…n. as shown in Equation (1). The Saaty’s nine point ratio scale attributes value for pairwise comparison where 1 represents “Equal Importance”, 3 represents “Moderate Importance”, 5 represents “Strong Importance”, 7 represents “Very Strong Importance” and 9 represents “Absolute/Extreme Importance”, while 2, 4, 6 and 8 are intermediate values within the above value scale.
(1)$$\begin{array}{l} \begin{array}{c} \;\;\;\;\;\;\;\;\;\;\;\;\;\;c_{1}\;\;\;\;\;\;\;\;c_{2}\;\;\;\;\;\;\;\;\;c_{3}\;\;\;\;\;\;\;..c_{{}_{n}} \end{array}\\ A=\left[\begin{array}{cccc} 1\; & a_{{}_{12}} & a_{13} & a_{1n}\\ {}^{1}/a_{12} & 1\; & a_{23} & a_{2n}\\ {}^{1}/a_{1n} & {}^{1}/a_{2n} & 1\; & a_{3n}\\ \vdots & \vdots & \vdots & \vdots \;\\ {}^{1}/a_{1n} & {}^{1}/a_{2n} & {}^{1}/a_{3n} & 1\; \end{array}\right]\begin{array}{c} A_{1}\\ A_{2}\\ A_{3}\\ \;\vdots \\ A_{n} \end{array} \end{array}$$

#### 2.2.2. Eigenvalue (*EV*)

After obtaining the PCM, The researchers normalize the value each row and average across the rows, then compute priority vector or the normalized principal eigenvector matrix of opinions where the EV is $w_{j}$ and $\lambda_{\max}$ is the maximum *EV*. A perfectly consistent criterion will result in $\lambda_{\max}=n$, as summarized in Equation (2).
(2)$$EV=\lambda_{\max}w$$

#### 2.2.3. Consistency Ratio Test (C.R.)

Then, a *C.R.* test is conducted to measure the degree of departure in the *EV* from pure inconsistency through the following steps. First, the researchers calculate the consistency index (*C.I.*) as shown in Equation (3), and then select the random consistency index (*R.I.*) varying to the size of n as shown in Table 2. Lastly, the researchers calculate the *C.R.* as shown in Equation (4) where the acceptable *C.R.* value is less than 0.1. Upon passing the criteria, the derived *EV* can be referred to as $w_{j}$. However, if *C.R.* is 0.1 or above, the weight of the IDCs must be adjusted or reassigned in the PCM. Let n be the number of IDCs, and the process can be summarized as follows.
(3)$$C.I.=\frac{\left(\lambda_{\max}-n\right)}{\left(n-1\right)}$$
(4)$$C.R.=\frac{C.I.}{R.I.}<0.1$$

### 2.3. The Technique for Order Preference by Similarity to Ideal Solution (TOPSIS)

TOPSIS, developed by Kwangsun Yoon and Hwang Ching-Lai [34], is suitable for multi-criteria decision making by assigning priority to ideal solutions. In the sustainability assessments, each IDC was measured and interpreted differently. For example, high values in some IDCs may indicate a higher level of significance, whereas in other groups of IDCs, low values can be interpreted as having higher level of significance. TOPSIS determines the IS value in both positive ideal (PI) and negative ideal (NI), then calculates the closest Euclidean distance to the PI as the best answer. To conduct TOPSIS analysis, it is necessary to monotonically increase quantitative data and the $w_{j}$ of each IDC. The process can be summarized as follows.

#### 2.3.1. Creation of Decision Matrix (DM) 

First, the researchers create a DM where the alternative (Alt) is the target city, ($x_{ij}$) is the data of each $Alt_{i}$ for each IDC*_j_* and $c_{i}$ is the IDC as summarized in Equation (5).
(5)$$\begin{array}{cc} & \;\;\;c_{1}\;\;\;\;\;\;c_{2}\;\;\;\;\cdots \;\;\;\;c_{i}\;\;\;\\ \begin{array}{c} Alt_{1}\\ Alt_{2}\\ \vdots \\ Alt_{m} \end{array} & \left[\begin{array}{cccc} x_{11} & x_{12} & \ldots & x_{1n}\\ x_{21} & x_{22} & \ldots & x_{2n}\\ \vdots & \vdots & \ddots & \vdots \\ x_{m1} & x_{m2} & \cdots & x_{mn} \end{array}\right] \end{array}$$

#### 2.3.2. Normalization of DM for Ease of Comparison

As each IDC has different units of measurement, it is necessary to convert $x_{ij}$ into a dimensionless or unitless index ($r_{ij}$) as calculated in Equation (6).
(6)$$r_{ij}\frac{x_{ij}}{\sqrt{\sum_{i=1}^{m}x_{ij}^{2}}}\;;\;\begin{array}{c} \forall i=1,\ldots ,m\\ \forall j=1,\ldots ,n \end{array}$$

#### 2.3.3. Weights of Normalized DM ($v_{ij}$)

This step is conducted by multiplying the $w_{j}$ in each IDC with the $r_{ij}$ derived from the normalized decision matrix in the previous step. This is summarized in Equation (7).
(7)$$v_{ij}=w_{j}r_{ij}\;v=\left[\begin{array}{cccc} v_{11} & v_{12} & \ldots & v_{1n}\\ v_{21} & v_{22} & \ldots & v_{2n}\\ \vdots & \vdots & \ddots & \vdots \\ v_{m1} & v_{m2} & \cdots & v_{mn} \end{array}\right]=\left[\begin{array}{cccc} w_{1}r_{11} & w_{1}r_{12} & \ldots & w_{n}r_{1n}\\ w_{1}r_{21} & w_{2}r_{22} & \ldots & w_{n}r_{2n}\\ \vdots & \vdots & \vdots & \vdots \\ w_{1}r_{m1} & w_{2}r_{m2} & \cdots & w_{n}r_{mn} \end{array}\right]$$

#### 2.3.4. Ideal Solution

As an IDC differently impacts decision-making, it is necessary to calculate both the positive ideal solution (*A+*), as illustrated in Equation (8), and the negative ideal solution (*A−*), as illustrated in in Equation (9). In Equations (8) and (9), J is the set of IDCs with a higher level of significance when reaching a higher value, and J′ is the set of IDCs with a higher level of significance when reaching a lower value. At this step, the reviewed HIS is used as a base to determine *A+* or *A−*.
(8)$$A+=\left\{v_{1}^{+},\ldots ,v_{n}^{+}\right\};\;\mathrm{where}\;v_{j}^{+}=\begin{array}{c} \max\left(v_{ij}\right)\mathrm{if}\;j\in J\;\\ \min\left(v_{ij}\right)\mathrm{if}\;j\in J\prime \end{array}$$
(9)$$A-=\left\{v_{1}^{-},\ldots ,v_{n}^{-}\right\};\;\mathrm{where}\;v_{j}^{-}=\begin{array}{c} \min\left(v_{ij}\right)\mathrm{if}\;j\in J\;\\ \max\left(v_{ij}\right)\mathrm{if}\;j\in J\prime \end{array}$$

#### 2.3.5. Deviation from Ideal Solutions

While a deviation from a positive ideal solution ($s_{i}^{+}$) can be calculated as shown in Equation (10), a deviation from a negative ideal solution ($s_{i}^{-}$) can be calculated per Equation (11). It is noted that regardless of the sequence, substituting $\left(v_{ij}-v_{j}^{+}\right)$ with $\left(v_{j}^{+}-v_{ij}\right)$ or substituting $\left(v_{ij}-v_{j}^{-}\right)$ with $\left(v_{j}^{-}-v_{ij}\right)$ results in the same value when squared.
(10)$$s_{i}^{+}=\sqrt{\sum_{j=1}^{n}{\left(v_{ij}-v_{j}^{+}\right)}^{2}}\forall i=1,\ldots ,m$$
(11)$$s_{i}^{-}=\sqrt{\sum_{j=1}^{n}{\left(v_{ij}-v_{j}^{-}\right)}^{2}}\forall i=1,\ldots ,m$$

#### 2.3.6. Relative Closeness ($c_{j}^{*}$)

Finally, Equation (12) is calculated for each $Alt_{i}$ where $Alt_{i}$ ranges between 0 and 1. The selection is least preferred when $c_{j}^{*}$ is close to zero. On the other hand, the selection is most preferred when $c_{j}^{*}$ is close to one.
(12)$$c_{j}^{*}\frac{s_{i}^{-}}{s_{i}^{+}+s_{i}^{-}}\;$$

## 3. Results

### 3.1. Criteria and Standard

Table 3 demonstrates sustainability in HEDm, consisting of four elements as follows:

Health status (HS) consists of five IDCs ranging from VR01–VR05. It was found that the world average life expectancy, or VR01, was 73.2 years, with the highest value found in Hong Kong at 85.29 years [35]. For the world average of low-birth-weight newborns, or VR02, the minimum value was 5% while the maximum value was over 20% [36]. Next, the world average death rate, or VR03, was 7.6 deaths per 1000 population [37]. The infant mortality livebirth rate, or VR04, was targeted to be 12 per 1000 livebirths at minimum and not to exceed 25 per 1000 livebirths by 2030 by SDGs [38]. Lastly, the world average suicide mortality rate, or VR05, was 5.6 deaths per 100,000 population at the minimum and 23.5 deaths per 100,000 population at the maximum [39].Communicable disease control (CDC) consists of four IDCs ranging from VR06–VR09. It was found that the current world average HIV/AIDS mortality rate, or VR06, was 11 deaths per 100,000 population with the expectation that the value will fall to 8.5 deaths per 100,000 population in 2040 [40]. Next, the world average tuberculosis mortality rate, or VR07, was less than 38 deaths per 100,000 population at the minimum and not more than 319 deaths per 100,000 population at the maximum [41]. For the world average pneumonia mortality rate, or VR08, the value was 34.31 deaths per 100,000 population, with the lowest number in Finland at the rate of 5.20 deaths per 100,000 population [42]. Lastly, the world average diarrhea mortality rate, or VR09, was 20.95 deaths per 100,000 population, with the lowest number in Montenegro at the rate of 0.09 deaths per 100,000 population [43].Non-communicable disease control (NCDs) consists of six IDCs ranging from VR10–VR15. It was found that the world average cancer mortality rate, or VR10, was 131.53 deaths per 100,000 population, with the lowest value in Kuwait at 71.26 deaths per 100,000 population [44]. Next, the world average stroke mortality rate, or VR11, was 84.19 deaths per 100,000 population, with the lowest value in Switzerland at a rate of 21.77 deaths per 100,000 population [45]. Then, the world average ischemic heart disease mortality rate, or VR12, mainly found in North Africa, the Middle East, Eastern Europe and Central Asia, was 112.37 deaths per 100,000 population [46]. For the world average diabetes mellitus mortality rate, or VR13, a report in 2019 found that the rate was 18.5 deaths per 100,000 population [47], while the world average chronic obstructive pulmonary disease mortality rate, or VR14, was 46.3 deaths per 100,000 population [48] and the world average kidney disease mortality rate, or VR15, was 77.01 deaths per 100,000 population [49].Health resources (HR) consists of seven IDCs ranging from VR16 to VR22. With references to the standards on healthcare resources set by the WHO based on national income, Thailand is categorized as an upper-middle-income country. The minimum ratio of physicians, or VR16, of the group was 10 physicians per 100,000 population, with the maximum ratio being 170 physicians per 100,000 population. Next, the minimum ratio of nursing and midwifery personnel, or VR18, was 60 persons per 100,000 population at the minimum, with a maximum ratio of 380 persons per 100,000 population. Then, the total health worker ratio, or VR20, was 210 workers per 100,000 population at the minimum, with a maximum ratio of 780 workers per 100,000 population [50]. For the current world average hospital beds ratio, or VR17, the value was 110 beds per 100,000 population [52] where the WHO has determined the benchmark should be at least 3 beds per 100,000 population [51]. The world average psychiatrist ratio, or VR19, was 1.7 psychiatrists per 100,000 population [54], whereas the recommended ratio for high-income countries was 6 psychiatrists per 100,000 population [53]. According to WHO standards, the ambulance ratio, or VR21, should be at least 1 ambulance per 100,000 population for people living in plains [56]. Nevertheless, the rate is set at 3.3 ambulances per 100,000 population for high-income countries [55]. Lastly, on the completion of electronic medical records, or VR22, the Ministry of Public Health (MOPH) of Thailand has published a policy enforcing that all electronic medical records must be at least 90% complete [57].

Table 4 demonstrates sustainability in ENDm, consisting of four elements as follows:

Environment risk management (ERM) consists of one IDC: VR23. MOPH has required each province to manage at least 60% of environmental risks involving one hygienic and environmental disease at minimum [58].Air pollution management (APM) consists of one IDC: VR24, which uses an average air quality index (AQI) reporting PM2.5, PM10, O_3_, CO, NO_2_ and SO_2_ in each city area as an element. AQI is divided into six levels from 0 to above 201, where an AQI value equal to 100 indicates standard air quality. When the AQI is above 100, the air pollution has exceeded the standard, thus affecting health [59,62].Protected Natural Areas (PNA) consists of one IDC: VR25. The Thai royal forest department has formulated its main strategy to preserve the prosperity of forests in Thailand by designating each city to have natural and artificial forest in at least 40% of the city area within 10 years [60].Water management (WM) consists of one IDC, VR26, by adopting a water management index (WMI) as an element. The index reports a score of water management in eight dimensions: (1) resources, (2) household water security, (3) economic water security, (4) balance in resources and usage, (5) environmental water security, (6) resilience to water-related disasters, (7) management of upstream forests and (8) water resource management performance. To quantify the evaluation, the index is assessed under five ranges where 0.00–1.00 means hazardous, 1.01–2.00 means engaged, 2.01–3.00 means capable, 3.01–4.00 means effective and 4.01–5.00 means model for water management [61].

Table 5 demonstrates sustainability in SODm, consisting of three elements as follows:

Health service standard (HSS) consists of six IDCs ranging from VR27 to VR32. Every IDC is defined by the policies from MOPH, which indicate the score for health resource management, or VR27, must be close to 0, not exceeding 4%. For transparency in public health, or VR28, all provincial health departments must pass the integrity and transparency assessment with a score equal to or more than 92%. With regards to green and clean hospital administration, or VR29, hospitals must organize green and clean activities and pass an assessment with a score of at least 98%. In addition, the management of public health crises, or VR30, must meet the criteria with an outstanding level of 100%, whereas the community hospital quality, or VR31, must meet Hospital Accreditation Level 3 with a minimum value of 90%. Lastly, the control of acute infectious diseases, or VR32, must be 100%.Social security (SS) consists of four IDCs ranging from VR33 to VR36. It was found that the world average smoking mortality rate, or VR33, was 95.61 deaths per 100,000 population, with the lowest mortality rate in Peru of 14.02 deaths per 100,000 population [63]. For the world average alcohol drinking mortality rate, or VR34, the value was 2.01 deaths per 100,000 population, with the lowest value found in Singapore at a rate of 0.16 deaths per 100,000 population [64]. In addition, the world average traffic accident mortality rate, or VR35, was 14.99 deaths per 100,000 population, with the lowest rate in Singapore of 2.64 deaths per 100,000 population [65]. Lastly, on the crime world average mortality rate, or VR36, the value was 5.4 deaths per 100,000 population, with the lowest rate found in Singapore of 0.4 deaths per 100,000 population [66].Health promotion (HP) consists of five IDCs ranging from VR37 to VR41. The IDCs in health promotion are also defined by the policies from MOPH, requiring that universal health coverage service, or VR37, desirable health behaviors, or VR38, management of glycemic control, or VR39, and management of blood pressure control, or VR40, must each be equal to or more than 80%, while the world average obesity (BMI > 30 kg/m^2^) or VR41 was 8.52 persons per 100,000 population [67].

Table 6 demonstrates sustainability in ECDm, consisting of four elements as follows:

City’s employment (CEM) consists of one IDC: VR42. To consider employment situation at a good level, each city should control its unemployment rate to be less than the current global average of 6.5% [68].Poverty reduction (PR) consists of one IDC: VR43. The average share of the population living in poverty was 8.44% [69].Household debt (HD) consists of one IDC: VR44. The appropriate debt bearing ratio was between 1% and 30%, not exceeding 43% [70].Economic growth (EGR) consists of one IDC: VR45. The Bank of Thailand predicted that the Thai economy would grow by 3.3% in 2019 [71].

### 3.2. Analyze Weight by Metric with AHP

Table 7 shows the analysis based on a pairwise comparison of each IDC pair within an element. The analysis found that every element had a C.R. value less than 0.1, and thus it can be concluded that the IDCs were appropriate and consistent enough to be used in the TOPSIS.

### 3.3. Application Research

The researchers decided to select Khon Kaen, a province in the northeastern part of Thailand, as a target of study because the province is a major population center of the northeastern region as shown in Figure 2. While the province is only 10,885 sq.km., the fifteenth largest province in Thailand, it is located 100–200 m above sea level with 25 districts covering 1,790,859 populations; it is therefore the fourth largest province in terms of population. Moreover, in terms of capacity, it is a center of education and technology with abundant natural resources and established infrastructure, including an airport and inland logistical hubs. Thus, it is promoted as a tourist destination, and it has also implemented a strategy to become a healthcare center in 2023. This study includes the assessment with TOPSIS analysis in all four dimensions: health, economic, environment and society. Based on the information disclosed by government agencies in 2019, as shown in Table A2, the data show consistencies in each dimension with the expected HIS values previously mentioned in Table 3, Table 4, Table 5 and Table 6. The researchers have followed the steps below to identify all 45 IDCs as shown in Table 8.

#### 3.3.1. Calculation of *A+* and *A−* for Each IDC

Firstly, the researchers arrange all IDC data with PI and NI values into DM per Equation (5). Then, the data is converted into a dimensionless or unitless index with the formula shown in Equation (6). With $w_{j}$ as derived in Table 7, the weighted normalized DM is computed with Equation (7). Lastly, calculate *A+* per the formula shown in Equation (8) and *A−* per the formula shown in Equation (9). The results are as shown in Table 9.

#### 3.3.2. Calculation of *S^+^* and *S^−^* for Each IDC

The researchers then evaluate whether each IDC is closer to *A+* or *A−*. When the value is closer to *A+*, it implies that the city can manage the targeted IDC to have few deviations from the HIS as listed in Table 3, Table 4, Table 5 and Table 6. In that case, the calculation is conducted per Equation (10). On the other hand, if the value is close to *A−*, the city needs to urgently improve the targeted IDC. The calculation in this case is conducted per Equation (11). The results are shown in Table 10.

#### 3.3.3. Calculation of Relative Closeness and Ranking of Each Element and Dimension

Relative closeness and ranking are calculated from *S^+^* and *S^−^* in Table 10 per Equation (12). At this step, the calculation is separately conducted for each factor in each dimension as previously mentioned in Table 3, Table 4, Table 5 and Table 6. The results are shown in Table 11.

### 3.4. Analysis and Strategies

Table 11 shows the overall sustainability of the case study province (CSP). The result has shown that economy ranks the highest of all sustainability dimensions in CSP, followed by society, then health and, finally, environment. Below, the researchers will only discuss the results with the lowest $c_{}^{*}$ in each dimension, as it is clearly indicated that there is deficiency in such dimensions that requires immediate improvement. The results are summarized in Figure 3.

From Figure 3, the results can be demonstrated for each dimension and element as follows.

#### 3.4.1. Environment

Air pollution management: The air pollution management has the lowest $c_{}^{*}$. Currently, CSP has moderate air quality that allows people to do outdoor activities, but the air quality is not suitable for members of sensitive groups. CSP should invest in technology to provide information and send real-time alerts on air conditions for people to be on guard in a timely manner.Protected natural areas: The Thai royal forest department has formulated a strategy for each city to have natural and artificial forests in at least 40% of their total area. At present, CSP has forested areas in only 11.91% of its total area with 4 national parks, 2 forest parks, 22 national forest reserves and 206 community forests. The forest biomes include hill evergreen forest, dry evergreen forest, mixed deciduous forest and dry deciduous dipterocarp forest. Within the forest are important plants and wild plants, which are invaluable natural resources. The plants categorized as important plants are Pradu, Makha Mong, Tabaek, Hieng, Antimony, Daeng, Teng and Rang while wild sprouts, wild galangal, peg, rattan, wild orchids, vetiver grass are categorized as wild plants. Due to high demand, the majority of inhabitants engage in agriculture, hunting and forestry, leading to issues in the demand on farming areas, forest degradation and forest encroachment. While forest degradation is caused by a lack of public understanding in related laws, forest encroachment is caused by harmful activities, such as illegal logging and clearing as well as slash and burn agriculture, without the awareness to preserve the forest. Moreover, the situation is worsened from insufficient wildfire-controlling equipment. As a result, forests are diminished every year, and wildfires also affect air conditions in the city.Authorities must raise awareness on the significance of forests and ecosystems by vigorously enforcing laws against intruders, educating about and clearly allocating farming area and land usage rights as well as formulating a policy to increase greenery, namely natural forests, urban forests or parks in the city. The city can also collaborate with universities to invest in sensor technology, such as heat detection to alert people of a wildfire occurrence. Forests also help reduce air pollution by photosynthesis, thus regaining environmental balance. In particular, seed-bearing plants can convert carbon monoxide into carbon dioxide, which is a necessary resource for photosynthesis; hence, it can be seen that forests can help reduce both carbon monoxide and carbon dioxide in the atmosphere.Water management: The CSP covers five river basins, namely the Churn River basin, Phong River basin, Chi River basin, Huai Aak River basin and Lum Phung Choo River basin. The most-utilized areas for agriculture and industry are in the Phong River basin and Chi River basin, respectively. Such usage leads to releases of industrial wastewater, especially in the Lam Nam Phong area where the water quality is critically damaged. Inspections have found that only 7% of water was in good quality, whereas another 53% was in fair condition and the other 40% was in a deteriorated state. Moreover, the quality of water during a drought is relatively deteriorated, affecting the livelihood of aquatic organisms as well as human consumption. Furthermore, contaminated water was also found, particularly in the Chi River flowing through Chonnabot district, Manchakhiri district and Muang district at the area of Tha Phra subdistrict, with high electrical conductivity caused by salinity leaching. Urban expansion also worsens water quality from the release of untreated wastewater into natural resources, as found in Huai Chik in Ban Phai district, Lam Huai Phra Khue district and Nong Khot swamp in Mueang district. With regards to Nam Phong River, water quality is deteriorated by two factors: wastewater from large factories producing pulp mills, sugar, liquor and alcohol and the expansion of fish farming. Water quality in drought is dramatically diminished in accordance with the WMI of CSP, which was found to be at a moderate level. Detailed study has found that an improved balance in resources and usage is needed urgently, and that the policymaker should plan for drought in summer and prepare for flood in rainy season. Previously, to solve the issue, the Kaem Ling Project was initiated by His Majesty Rama IX to address the flooding problem in Thailand and prevent the intrusion of saltwater during drought into rivers, canals and agricultural areas. As the project stores freshwater above floodgates, people can utilize water for agriculture, industrial usage and consumption. At the same time, the project also indirectly helps preserve water and the environment, as the water from reservoirs dilutes sewage when released into ditches and pushes effluent into the sea.Environment risk management: Even environmental elements were assessed with the highest $c_{}^{*}$. When comparing with the standards set by MOPH that each province must have ERM more than 60%, CSP was found to have attained only 25%. As a result, responsible agencies should implement measures as follows. (1) Basic level: launch a measure to accurately identify environmental issues affecting health within the area together with an action plan at the local level. (2) Intermediate level: establish a working group at the provincial level to manage environmental problems and a team of specialists to investigate occupational and environmental diseases with at least one team per province. In addition, the team should manage environmental problems concretely affecting health as well as emphasizing at least two environmental elements for health promotion. (3) Advanced level: the city should be able to accommodate 50% of the population in relation to risk management and health protection.

#### 3.4.2. Health

Health resource: The assessment has shown that the health resource element has the lowest $c_{}^{*}$ value. CSP is equipped with the most complete and abundant public health resources in northeastern Thailand. Such resources include 32 hospitals which can be categorized by their lines of management as follows: 23 hospitals under provincial public health offices, 2 university hospitals in Khon Kaen province, 1 hospital under the Ministry of Defense, 1 hospital under the Department of Health, 1 hospital under the Department of Mental Health, 1 hospital under the medical department and 3 private hospitals. All of these hospitals, with 248 subdistrict health-promoting hospitals, are responsible for caring for the people living in the area. While CSP is geared to be a national center of excellence for medical services in the future—or Medicopolis, as defined by the Thailand Center of Excellence for Life Sciences (TCELS)—with efforts to create an ecosystem among academies, researchers and startups with a regulatory sandbox in the area, the present capability of public health resources, as discussed in VR16–VR20, is under international standards. Hence, CSPs should improve those IDCs to meet the criteria stipulated by the WHO, such as the requirement for physician ratios to be between the minimum of 10 physicians and the maximum of 170 physicians per 100,000 population, the hospital bed ratio to be between 110 beds at minimum and 300 beds at maximum per 100,000 population, the number of nursing and midwifery personnel to be between 60 persons at minimum and 380 persons at maximum per 100,000 population, the number of psychiatrists to be between 1.7 persons at minimum and 6 persons per 100,000 population at maximum and the total health workers to be between 210 persons at minimum and 780 persons at maximum per 100,000 population to achieve the target of being a holistic medical center in the future.Non-communicable disease control: It is found that management of all IDCs in CSP is lower than the HIS value shown in Table 2 except VR13, or diabetes mellitus mortality rate. Diabetes is caused by many factors, including genetic factors and environmental factors with increased risk for people over 45 years, people with genetic inheritances, Asians with a body mass index (BMI) from 25 to 23, people with high blood pressure of 140/90 mm Hg and people who are physically inactive. MOPH should provide more information on self-monitoring, preliminary diabetes risk assessment and the communal level of screening. To detect the issue early, it is necessary to coordinate with village health volunteers and medical volunteers on screenings to access the at-risk groups and register them into the healthcare system to control the severity of the illness before it induces other complications.Communicable disease control: All IDCs were found to be lower than the HIS values listed in Table 2 except VR08, or pneumonia mortality rate. Pneumonia can occur within the general public, especially in young children under two years old and people over 65 years old. The disease is also found in hospitalized patients, especially in the intensive care unit (ICU) or for people with chronic diseases such as diabetes, asthma, chronic obstructive pulmonary disease and heart disease as well as patients who are immunocompromised, such as HIV-infected or AIDS patients, organ transplant recipients, patients receiving chemotherapy or taking immunosuppressants for a long period. To address the issue, the MOPH must allocate vaccination to these groups, including young children, people over 65 years old and people with low immunity or certain congenital diseases. Treatments include providing flu vaccination to reduce occurrences of infectious pneumonia, pneumococcal vaccine to prevent infection in adults and streptococcus pneumonia vaccine to prevent infection in children.

#### 3.4.3. Social

Health service standard: The results show that all IDCs had attained a value above HIS except in VR28, or transparency in public health, and VR32, or control of acute infectious diseases. The units under the Office of the Permanent Secretary for Public Health that did not pass the Integrity and Transparency Assessment (ITA) must be inspected and have the following requirements enforced: (1) periodically disclose active procurement information to the public on websites to ensure transparency in procurement each month and fiscal year; (2) implement policies on human resource management and development with respect to human rights and dignity; (3) provide channels for complaints with statistical reports of said complaints; (4) formulate a preventive action plan on anti-corruption, anti-misconduct and ethics promotion with reports of activities not in compliance with the code of ethics; (5) stimulate measures to prevent and solve issues on harassment, including sexual harassment in the workplace.Social safety: All IDCs were found to be above HIS except VR34, or alcohol drinking mortality, and VR35, or traffic accident mortality. Based on the result, the MOPH should establish a unit properly providing guidance on ending alcoholism. The unit should be privately and easily accessed, and counselees should be able to be regarded as anonymous. To address the problem in VR35, infrastructures should be taken into consideration, as CSP is a transportation hub with the highest population in the country and an increasing number of cars every year regardless of the fact that infrastructure has not been upgraded or improved. As a result, existing infrastructure cannot sufficiently support rising numbers of vehicles, thus causing road accidents. The government should assign importance to infrastructure projects for accident prevention such as (1) installation of flexible posts instead of steel poles to reduce impulse of force; (2) installation of road studs to visibly show lane lines at night; (3) provision of speed humps to slow down vehicles, especially in crowded areas; (4) installation of curved glass to reduce accidents in blind spots; (5) installation of solar street lights that can function even in electricity-inaccessible areas and curved warning lights to increase visibility both during the day and at night; (6) installation of Euro Smart crash cushions to absorb forces from collisions at speeds of 50–110 km/h in dangerous areas such as traffic islands, merge lanes and the beginnings and ends of bridges; (7) provision of safety islands in areas with heavy traffic to raise safety and reduce road accidents; (8) installation of overpasses, U-turn bridges and smart traffic lights.Health promotion: The assessment has shown that all IDCs are above HIS except VR40, or management of glycemic control, and VR41, or management of blood pressure control. The result is consistent with causes from factors in NCDs that cause the diabetes mortality rate to exceed the world average in CSP. VR40 and VR41 are important tools for screening people at risk of diabetes and high blood pressure so they may register and receive proper treatment. Responsible agencies should accelerate the screening as much as possible to reduce the mortality rate caused by the disease by first seeking cooperation with medical volunteers to hold preliminary screenings for the groups and register them into the system, and then conducting detailed examinations at hospitals to properly treat these groups before the disease becomes more severe.

#### 3.4.4. Economic

The assessment results have found that both IDCs—VR 42, or unemployment rate, and VR43, or population living in poverty—are up to standard. The result is consistent with the location of CSP, as it situated in the East–West Economic Corridor at the junction of north–south economic links to the eastern seaboard. Here, the government is promoting the province to be one of three pilot provinces for the development of information and communication technology with potential for trade and investment, as the city is the center of the northeastern region with readiness in its infrastructure.

Household debt: the assessment has found that the value exceeds the standard, which is in line with the CPS data that the number of households with low incomes is increasing annually. Hence, the government should provide assistance with debt reduction as follows: (1) improve repayment methods on education loans, reordering the order of debt written-off and lowering the default interest rate; (2) take debt restructuring as its main agenda to reduce the burdens of debtors; (3) solve issues around hire purchase debt of automotive by appropriately supervising fair conduct in vehicle leasing businesses; (4) solve debt problems of civil servants; (5) reduce and review structures and ceilings of interest rates and fees to help people and SMEs access credit at an appropriate rate; (6) implement the BOT’s Debt Clinic project to encourage the conversion of nonmortgage debt from credit cards, cash cards and personal loans into long-term credit with low interest; (7) improve judicial processes to facilitate debt litigation.Economic growth: Despite the assessment result that economic growth has the highest $c_{}^{*}$, the growth rate, which considered to be acceptable, is still lower than expected, which is mainly due to the pandemic at the end of the year. The government should take measures to preliminarily stimulate the economy, especially after the pandemic, as CSP is a province with high economic sustainability because of its location in the East–West Economic Corridor and at the junction of the north–south economic links to the eastern seaboard. When considering gross provincial product (GPP), the main income of CSP largely comes from the nonagricultural sector, contributing 88.37% of the total GPP. Detailed statistics show that 36.56% comes from manufacturing, while services account for 49.48% of the total GPP. Within the service sector, educational services account for 11.34%, while trade and automotives account for 11.27%. Hence, it can be concluded that CSP has outstanding capabilities in the manufacturing, education, trade and automotive sectors consistent with its location close to the cross-border trade area in Mukdahan province with abundant opportunity to export products to neighboring countries.

The above analysis can be summarized into strategies as Figure 4 shows. With reference to the Kano house model, the top of the house is the target of CSP aiming to be a sustainable city for better quality of life and wellbeing. The two-level foundation consists of a Plan–Do–Check–Act cycle on health indicator standards and supporting technology for sustainability development. The goal is supported by four pillars of sustainability: health sustainability, environmental sustainability, social sustainability and economic sustainability. Based on the analysis, the researchers provide three levels of strategies: growth maintaining strategy, monitoring strategy and improvement strategy with the details described as below.

Level 1: Growth-maintaining strategy, which is applied to maintain the current performance of the IDCs that are already above the standard with a target to achieve a higher value of the IDC in the future. The strategy applies to the following:The first pillar, or health sustainability: As the analysis has found a number of areas with room for improvement, the elements applicable to this strategy include VR03, or number of deaths, VR04, or infant mortality livebirth, VR06, or HIV/AIDS mortality, VR07, or Tuberculosis mortality, VR12, or ischemic heart disease mortality, VR14, or chronic obstructive pulmonary disease mortality, VR15, or kidney disease mortality, VR21, or ambulance, and VR22, or electronic medical records.The third pillar, or social sustainability: three elements are applicable to this strategy, which are VR29, or green and clean hospital administration, VR30, or management of public health crises, and VR31, or community hospital quality.The fourth pillar, or economic sustainability: only VR42, or unemployment rate, is applicable to the strategy.

Level 2: Monitoring strategy, which is applied to monitor the IDCs whose performances are at the standard level. During the observation, different levels of strategy can be applied. If the IDCs attain better value, then the Level 1 strategy will be applied; on the other hand, if the IDCs are not performing, the strategy will alert observers to take precautionary measures to prevent the IDCS from falling to the level needed to apply the Level 3 strategy. This strategy applies to the following:The first pillar, or health sustainability: there are five elements that are applicable to this strategy, including VR01, or life expectancy, VR02, or low-birth-weight newborns, VR05, or suicide mortality, VR09, or diarrhea mortality, and VR11, or stroke mortality.The third pillar, or social sustainability: four elements are applicable to this strategy, including VR32, or control of acute infectious diseases, VR33, or smoking mortality rate, VR36, or crime mortality rate, and VR39, or obesity (BMI > 30 kg/m^2^).The fourth pillar, or economic sustainability: only VR43, or population living in poverty, is applicable to this strategy.

Level 3: Improvement strategy, which is applied to IDCs with a value below the standard. Hence, the IDCs must be urgently improved, and the CSP must set these as urgent matters by formulating a strategy as well as implementing technologies as solutions to improve the value of these IDCs to the point that they will be able to switch to Level 1 and Level 2 strategies. This strategy applies to the following:The first pillar, or health sustainability: there are seven elements applicable to this strategy, including VR08, or pneumonia mortality, VR13, or diabetes mellitus mortality, VR16, or physicians per 100,000 population, VR17, or hospital beds per 100,000 population, VR18, or nursing and midwifery personnel per 100,000 population, VR19, or psychiatrists per 100,000 population, and VR20, or total health workers per 100,000 population.The second pillar, or environmental sustainability: only four elements are applicable to this strategy. The elements are VR23, or environmental risk management, VR24, or average of AQI index, VR25, or forest area rate, and VR26, or water management index.The third pillar, or social sustainability: there are six elements applicable to this strategy. The elements are VR27, or health resource management, VR28, or transparency in public health, VR34, or alcohol drinking mortality rate, VR35, or traffic accident mortality rate, VR40, or management of glycemic control, and VR41, or management of blood pressure control.The fourth pillar, or economic sustainability: only VR44, or household debt per income ratio, and VR45, or gross provincial product growth rate, are applicable to this strategy.

## 4. Discussion

This study presents (1) a postpandemic urban sustainability assessment index in 4 dimensions and 15 elements with 45 indicators on health, environment, society and economy to comprehensively assess postepidemic sustainable development. The results were used to identify the strengths and weaknesses of cities in each dimension together with elements to develop and improve each city towards sustainability in the future. (2) benchmarks on the world average and targeted values of all 45 indicators from reliable international agencies with references on criteria to calculate A+ and A− in TOPSIS analysis. The references support the assessment by reflecting actual conditions as much as possible. (3) TOPSIS analysis with references to targeted benchmarks. By referring to standard value, comparison with other cities that may not reflect the actual situation is no longer necessary. Moreover, it improves the assessing mechanism, as the existing method to calculate A+ and A− by using data from cities with the highest targeted values as a calculation base with the relative closeness equal to one may not be the best value when compared with the targeted benchmark. Regarding the objective, the research was not intended to state which cities are the most sustainable, but to answer in which dimensions are the case study city the most sustainable and which elements should be improved to meet the benchmark. (4) TOPSIS analysis of the CSP in Thailand with one CSP driving to be a center of excellence in healthcare. The result has found that that CSP has the highest sustainability index in economy. When considered by element, the CEM had the highest impact while the HDD had the lowest impact. The second highest sustainability index was in the society dimension where HP had the highest impact and HSS had the lowest impact. In the health dimension, CDC was the most significant while HR played a minimal role. Lastly, in the environmental dimension, ERM played the most significant role while APM showed the least effect. The results can be interpreted to indicate that when an IDC achieves the highest value, CSP is performing well. In contrast, a low value in an IDC signals CSP to improve to meet the standard. Overall, the results can be utilized as policy recommendation to authorities for decision-making on accurate and efficient budgeting for sustainability development.

### Limitations and Future Research

While the research has served its purpose in identifying dimensions to be developed, the researcher recognize that it has encountered a number of limitations that must be addressed for further improvement in future research.

Although 15 experts providing opinions are suitable for using AHP analysis in this research, future research may require a larger number of experts from different fields such as city development, economy, social development and environment to gather more opinions from multiple aspects.This research was conducted at provincial level, while future research can extend the scope to a regional or national level to identify and address issues on a macro level.Since this research uses HIS values as a basis of comparison, it is necessary to update data to reflect the latest conditions.While this research focuses on assessing and prioritizing indicators as well as the city in their current conditions, it does not indicate precedent conditions or consequences.Future research should incorporate other analyses such as regression analysis to compare and identify causes and effects of each indicator that may change in the future.

## 5. Conclusions

The results of this research provide a sustainability assessment model that integrates the health dimension into the existing three dimensions of sustainable development with clear quantitative indicators in each component for a city. It also develops a method for TOPSIS analysis compared with HISes from reliable organizations. As TOPSIS is a dynamic model, it can be applied to urban and local development and assess actual conditions of cities. Advanced utilization of this study is demonstrated below.

Health indicator standards from this research can be used to establish goals for urban development in various dimensions of sustainable city development to comply with international standards.The IDC weight obtained from the AHP technique can be used to determine the level of importance of each IDC. It may be determined from the IDC results which city has the potential to conduct an assessment first. The IDCs are not necessarily to be conducted concurrently, as it may affect the efficiency of budget management of the city. On the other hand, it is also possible to adjust IDCs and HISes to fit with secondary data of the city before usage; even so, such adjustments should be within the objectives.At present, local leaders in Thailand often make decisions based on personal opinions, thus resulting in a lack of information on the real problems and urgent needs of the city. The result is inefficient investments that may not match actual expectations at the time.This research shall enable policymakers and decision-makers in budget planning at both at the national and local level to assess and select strategies for urban development. With indicators in 4 dimensions and 15 elements, weaknesses and strengths are shown and utilized in budget planning to precisely develop sustainability from the health aspect, leading to efficient and effective budget utilization and tremendous improvement in the quality of life of the people in the city.

## Figures and Tables

**Figure 1 ijerph-20-00259-f001:**
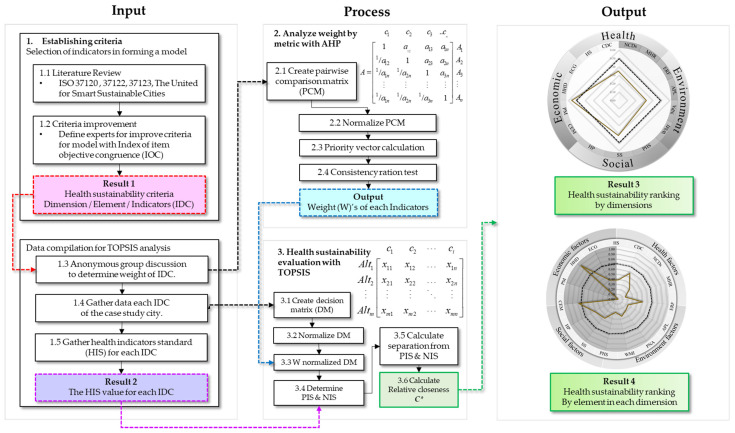
The conceptual framework.

**Figure 2 ijerph-20-00259-f002:**
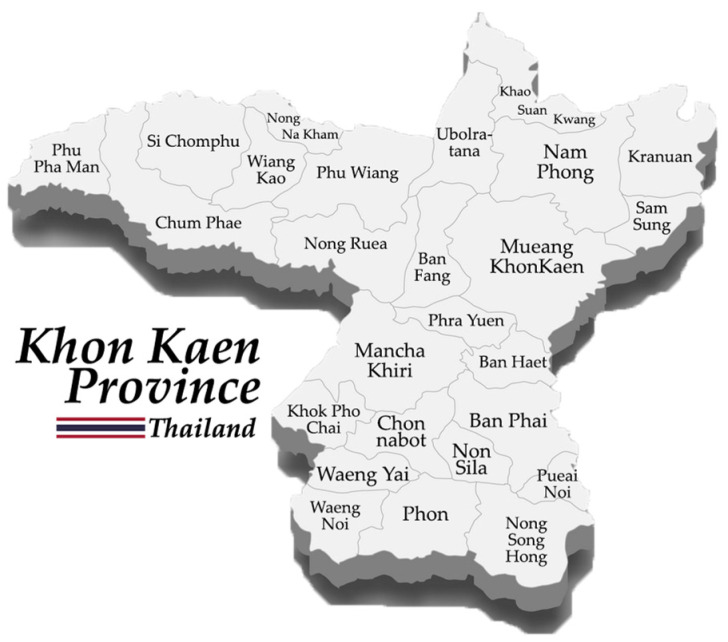
Map of Khon Kaen province, Thailand.

**Figure 3 ijerph-20-00259-f003:**
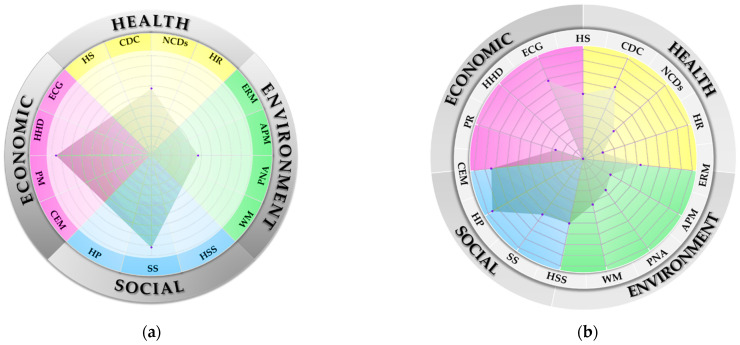
Sustainability assessment (**a**) by overall dimension and (**b**) by element in each dimension.

**Figure 4 ijerph-20-00259-f004:**
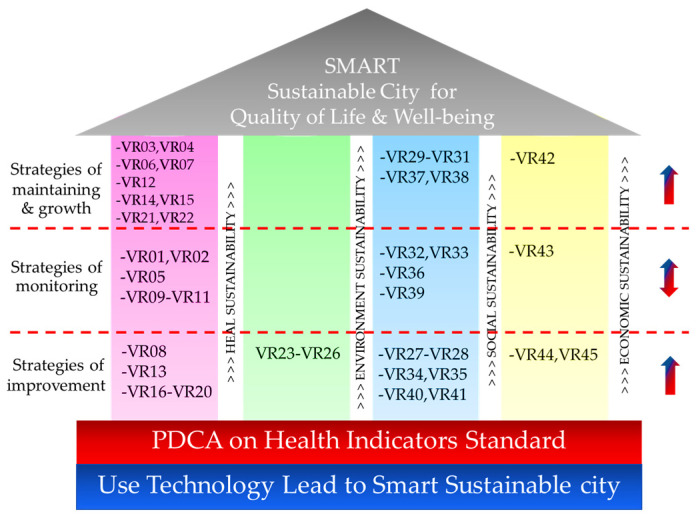
Three levels for the city’s improvement strategies.

**Table 1 ijerph-20-00259-t001:** Literature reviews matrix.

Objective	DM/NOC	NOA	ET	WT	CW	LOA			Ref
						DM	EM	RK	
Evaluate the level of sustainable development with the phenomenon of spatial clustering: a case study in China	Env. = 7	287	TOPSIS	ETP	Alt	3			[2]
	Soc. = 10								
	Econ. = 6								
Evaluate urban sustainability where grey relation analysis is used to reduce uncertainty in the process of evaluation: a case study in China	Env. = 12	16	TOPSIS	ETP	Alt	3			[18]
	Soc. = 15								
	Econ. = 12								
Assess sustainable urban development in an emerging economy with fuzzy-TOPSIS where there is a lack of clear data: a case study in Vietnam	Env. = 7	4	TOPSIS	Fuzzy-AHP	Alt	3			[19]
	Soc. = 7								
	Econ. = 6								
Assess a healthy and safe built environment with integrated MCDM methods: a case study in Lithuania	Env. = 4	21	MCDM	EP	Alt	3			[17]
	Soc. = 5								
	Econ. = 5								
Build a framework to assess the relative performances of multiple neighborhood renewal projects through a hybrid AHP-TOPSIS method: a case study in China	Env. = 6	18	TOPSIS	EP	Alt	1		✓	[20]
	Soc. = 3			-AHP					
	Econ. = 2								
	Etc. = 2								
Assess a sustainability assessment method that integrates the MCDM approach with the variability of the alternatives’ performance measurements: a case study in Europe	Env. = 4	26	TOPSIS	CRITIC	Alt			✓	[16]
	Soc. = 3								
	Econ. = 3								
Verify the performance of Brazilian municipalities in three dimensions of sustainability: a case study in Brazil	Env. = 5	217	TOPSIS	-	Alt	3	4	✓	[21]
	Soc. = 20								
	Econ. = 6								
Develop an evaluation index system that satisfies the requirements of green development in coal resource-based cities by considering four dimensions: a case study in China	Env. = 9	30	TOPSIS	ETP	Alt	4		✓	[22]
	Soc. = 9			-AHP					
	Econ. = 5								
	Etc. = 6								
Evaluate the progress of a single city towards the concept of sustainable development: a case study in Poland	Env. = 14	1	TOPSIS	DMK	YS	3		✓	[5]
	Soc. = 18								
	Econ. = 16								
Present a new model called Geo Umbria SUIT that integrates multicriteria analysis and geographic information systems and is specifically developed to help decision-makers make policy decisions about sustainability in planning: a case study in Malta	Env. = 6	6	TOPSIS	DMK	Alt	3		✓	[23]
	Soc. = 6								
	Econ. = 6								
Test the Geo Umbria SUIT model, found very suitable for territorial sustainability assessment, for evaluating sustainability at the territorial level of two different European countries: Italy and Spain	Env. = 6	17/20	TOPSIS	DMK	Alt	3		✓	[24]
	Soc. = 6								
	Econ. = 6								
Assessment of localizations in the Besancon area in terms of sustainable urban development: a case study in France	Env. = 3	8	SAW	Equal	Alt	3	3	✓	[25]
	Soc. = 3								
	Econ. = 3								
Assessment of sustainable socioeconomic development in European Union countries	Soc. = 4	13	EDAS	-	Alt			✓	[26]
	Econ. = 9								
Evaluate the sustainability of four dimensions towards the concept of sustainable development with the Dynamic Evaluation Model of Health Sustainability Under MCDM Benchmarking Health Indicator Standard: a case study in Thailand	Health = 4	Single or more	TOPSIS	AHP	HIS	4	15	✓	This paper
	Env. = 4								
	Soc. = 3								
	Econ. = 4								

Note: (NOC = Dimension-Number of criteria) Heal. = health/Env. = Environment/Soc. = social/Econ. = economic/etc.; (NOA = Number of alternatives)/(ET = Evaluation Technique); (LOA = Level of Assessment) DM = Dimension/Element = EM/Ranking = RK; (WT = Weight technique) ETP = Entropy/DMK = Decision maker’s/EP = Expert; (CW = Compare with) Alt = alternative/HIS = Health indicator standard/YS = Year series.

**Table 2 ijerph-20-00259-t002:** Random Consistency Index.

n	1	2	3	4	5	6	7	8	9	10	11	12	13	14	15
R.I.	0.0	0.0	0.58	0.90	1.12	1.24	1.32	1.41	1.45	1.49	1.51	1.48	1.56	1.57	1.59

**Table 3 ijerph-20-00259-t003:** Criteria and standards for HEDm.

Dimension	Element	IDC	Unit	OD	PI	NI	CODE	Ref
Health	Health Status	Life expectancy	Year	MAX	85.29	73.2	VR01	[35]
		Low-birth-weight newborns	%	MIN	5	20	VR02	[36]
		Number of deaths	1K	MIN	-	7.6	VR03	[37]
		Infant mortality livebirth	1K	MIN	12	25	VR04	[38]
		Suicide mortality	100K	MIN	5.60	23.5	VR05	[39]
	Communicable Disease Control	HIV/AIDS mortality	100K	MIN	8.5	11	VR06	[40]
		Tuberculosis mortality	100K	MIN	38	319	VR07	[41]
		Pneumonia mortality	100K	MIN	5.20	34.31	VR08	[42]
		Diarrhea mortality	100K	MIN	0.09	20.95	VR09	[43]
	Non-communicable Disease Control	Cancer mortality	100K	MIN	71.26	131.53	VR10	[44]
		Stroke mortality	100K	MIN	21.77	308.08	VR11	[45]
		Ischemic heart disease Mortality	100K	MIN	-	112.37	VR12	[46]
		Diabetes mellitus mortality	100K	MIN	-	18.5	VR13	[47]
		Chronic obstructive pulmonary disease mortality	100K	MIN	-	46.3	VR14	[48]
		Kidney disease mortality	100K	MIN	-	77.01	VR15	[49]
	Health resource	Physicians	100K	MAX	170	10	VR16	[50]
		Hospital beds	100K	MAX	300	110	VR17	[51,52]
		Nursing and midwifery personnel	100K	MAX	380	60	VR18	[50]
		Psychiatrist	100K	MAX	6	1.7	VR19	[53,54]
		Total Health worker	100K	MAX	780	210	VR20	[50]
		Ambulance	100K	MAX	3.3	1	VR21	[55,56]
		Electronic Medical Records	%	MAX	100	90	VR22	[57]

Note: 1K = per 1000 Populations.

**Table 4 ijerph-20-00259-t004:** Criteria and standard for ENDm.

Dimension	Element	IDC	Unit	OD	PI	NI	CODE	Ref
Environment	Environment risk management	Environmental risk management	%	MAX	60	-	VR23	[58]
	Air pollution management	Average of AQI index	Index	MIN	1	100	VR24	[59]
	Protected natural areas	Forest area rate	%	MAX	40	-	VR25	[60]
	Water management	Water management index	Index	MAX	5	2	VR26	[61]

**Table 5 ijerph-20-00259-t005:** Criteria and standard for SODm.

Dimension	Element	IDC	Unit	OD	PI	NI	CODE	Ref
Social	Health Service Standard	Health resource management	%	MIN	1	4	VR27	[58]
		Transparency in public health	%	MAX	92	-	VR28	[58]
		Green and clean hospital administration	%	MAX	98	-	VR29	[58]
		Management of public health crises	%	MAX	100	-	VR30	[58]
		Community hospital quality	%	MAX	90	-	VR31	[58]
		Control of acute infectious diseases	%	MAX	100	-	VR32	[58]
	Social security	Smoking mortality	100K	MIN	14.02	95.61	VR33	[63]
		Alcohol drinking mortality	100K	MIN	0.16	2.01	VR34	[64]
		Traffic accident mortality	100K	MIN	2.64	14.99	VR35	[65]
		Crime mortality	100K	MIN	0.4	5.4	VR36	[66]
	Health Promotion	Universal health coverage Service	%	MAX	80	-	VR37	[58]
		Desirable health behaviors	%	MAX	80	-	VR38	[57]
		Obesity (BMI > 30 kg/m^2^)	%	MIN	1.82	8.52	VR39	[67]
		Management of glycemic control	%	MAX	80	-	VR40	[57]
		Management of blood pressure control	%	MAX	80	-	VR41	[57]

Note: 1K = per 1000 Populations.

**Table 6 ijerph-20-00259-t006:** Criteria and standard for ECDm.

Dimension	Element	IDC	Unit	OD	PI	NI	CODE	Ref
Economic	City’s employment	Unemployment rate	%	MIN	-	6.5	VR42	[68]
	Poverty reduction	Population living in poverty	%	MIN	-	8.44	VR43	[69]
	Household debt	Household debt per income ratio	%	MIN	-	43	VR44	[70]
	Economic growth	Gross provincial product growth rate	%	MAX	3.3	-	VR45	[71]

**Table 7 ijerph-20-00259-t007:** Weight analysis of each IDC with AHP.

Pairwise Comparison Matrix (PCM)								Normalized-Pairwise Comparison Matrix (NPCM)								
CODE	VR01	VR02	VR03	VR04	VR05			CODE	VR01	VR02	VR03	VR04	VR05	SUM (EV)		
VR01	1	5	3	5	4			VR01	0.50	0.50	0.56	0.45	0.40	0.48		
VR02	1/5	1	1/3	2	2			VR02	0.10	0.10	0.06	0.18	0.20	0.13		
VR03	1/3	3	1	2	2			VR03	0.17	0.30	0.19	0.18	0.20	0.21		
VR04	1/5	½	½	1	1			VR04	0.10	0.05	0.09	0.09	0.10	0.09		
VR05	¼	½	½	1	1			VR05	0.13	0.05	0.09	0.09	0.10	0.09		
SUM	1.98	10	5.33	11	10			SUM	1	1	1	1	1	1		
PCM: Health status (HS)								NPCM: HS/N = 5, C.I. = 0.06, R.I. = 1.12, C.R. = 0.04 < 0.1								
CODE	VR06	VR07	VR08	VR09				CODE	VR06	VR07	VR08	VR09	SUM(EV)			
VR06	1	3	4	5				VR06	0.56	0.65	0.47	0.42	0.53			
VR07	1/3	1	3	4				VR07	0.19	0.22	0.35	0.33	0.27			
VR08	¼	1/3	1	2				VR08	0.14	0.07	0.12	0.17	0.12			
VR09	1/5	¼	½	1				VR09	0.11	0.05	0.06	0.08	0.08			
SUM	1.78	4.58	8.50	12.00				SUM	1	1	1	1	1			
PCM: Communicable disease control (CDC)								NPCM: CDC/N = 4, C.I. = 0.06, R.I. = 0.90, C.R. = 0.04 < 0.1								
CODE	VR10	VR11	VR12	VR13	VR14	VR15		CODE	VR10	VR11	VR12	VR13	VR14	VR15	SUM (EV)	
VR10	1	5	3	1	5	5		VR10	0.34	0.31	0.38	0.34	0.31	0.31	0.33	
VR11	1/5	1	1/3	1/5	1	1		VR11	0.07	0.06	0.04	0.07	0.06	0.06	0.06	
VR12	1/3	3	1	1/3	3	3		VR12	0.11	0.19	0.13	0.11	0.19	0.19	0.15	
VR13	1	5	3	1	5	5		VR13	0.34	0.31	0.38	0.34	0.31	0.31	0.33	
VR14	1/5	1	1/3	1/5	1	1		VR14	0.07	0.06	0.04	0.07	0.06	0.06	0.06	
VR15	1/5	1	1/3	1/5	1	1		VR15	0.07	0.06	0.04	0.07	0.06	0.06	0.06	
SUM	2.93	16.00	8.00	2.93	16.00	16.00		SUM	1	1	1	1	1	1	1	
PCM: Non-communicable disease control (NCDs)								NPCM: NCDs/N = 6, C.I. = 0.02, R.I. = 1.24, C.R. = 0.01< 0.1								
CODE	VR16	VR17	VR18	VR19	VR20	VR21	VR22	CODE	VR16	VR17	VR18	VR19	VR20	VR21	VR22	SUM (EV)
VR16	1	4	3	1	5	5	3	VR16	0.30	0.38	0.28	0.29	0.25	0.22	0.32	0.29
VR17	¼	1	1	1/3	3	5	1	VR17	0.08	0.09	0.09	0.10	0.15	0.22	0.11	0.12
VR18	1/3	1	1	1/3	3	3	0.5	VR18	0.10	0.09	0.09	0.10	0.15	0.13	0.05	0.10
VR19	1	3	3	1	5	5	3	VR19	0.30	0.28	0.28	0.29	0.25	0.22	0.32	0.28
VR20	1/5	1/3	1/3	1/5	1	1	0.5	VR20	0.06	0.03	0.03	0.06	0.05	0.04	0.05	0.05
VR21	1/5	1/5	1/3	1/5	1	1	0.33	VR21	0.06	0.02	0.03	0.06	0.05	0.04	0.04	0.04
VR22	1/3	1	2	1/3	2	3	1	VR22	0.10	0.09	0.19	0.10	0.10	0.13	0.11	0.12
SUM	3.32	10.53	10.67	3.40	20.00	23.00	9.33	SUM	1	1	1	1	1	1	1	1
PCM: Health resource (HR)								NPCM: HR/N = 7, C.I. = 0.02, R.I. = 0.90, C.R. = 0.02 < 0.1								
CODE	VR23	VR24	VR25	VR26				CODE	VR23	VR24	VR25	VR26	SUM (EV)			
VR23	1	1/3	1/5	1/3				VR23	0.08	0.06	0.10	0.08	0.08			
VR24	3	1	1/3	1				VR24	0.25	0.19	0.16	0.23	0.21			
VR25	5	3	1	2				VR25	0.42	0.56	0.49	0.46	0.48			
VR26	3	1	½	1				VR26	0.25	0.19	0.25	0.23	0.23			
SUM	12.00	5.33	2.03	4.33				SUM	1	1	1	1	1			
PCM: Environment dimension (ENDm)								NPCM: ENDm/N = 4, C.I. = 0.05, R.I. = 1.32, C.R. = 0.01 < 0.1								
CODE	VR27	VR28	VR29	VR30	VR31	VR32		CODE	VR27	VR28	VR29	VR30	VR31	VR32	SUM (EV)	
VR27	1	3	5	5	4	4		VR27	0.45	0.59	0.29	0.29	0.41	0.40	0.41	
VR28	1/3	1	5	5	3	3		VR28	0.15	0.20	0.29	0.29	0.31	0.30	0.26	
VR29	1/5	1/5	1	1	1/3	½		VR29	0.09	0.04	0.06	0.06	0.03	0.05	0.06	
VR30	1/5	1/5	1	1	1/3	½		VR30	0.09	0.04	0.06	0.06	0.03	0.05	0.06	
VR31	¼	1/3	3	3	1	1		VR31	0.11	0.07	0.18	0.18	0.10	0.10	0.12	
VR32	¼	1/3	2	2	1	1		VR32	0.11	0.07	0.12	0.12	0.10	0.10	0.10	
SUM	2.23	5.07	17	17	9.67	10		SUM = 1	1	1	1	1	1	1	1	
PCM: Health service standard (HSS)								NPCM: HSS/N = 6, C.I. = 0.06, R.I. = 1.24, C.R. = 0.03 < 0.1								
CODE	VR33	VR34	VR35	VR36				CODE	VR33	VR34	VR35	VR36	SUM (EV)			
VR33	1	1/3	1/5	1/5				VR33	0.07	0.05	0.08	0.08	0.07			
VR34	3	1	1/3	1/3				VR34	0.21	0.14	0.13	0.13	0.15			
VR35	5	3	1	1				VR35	0.36	0.41	0.39	0.39	0.39			
VR36	5	3	1	1				VR36	0.36	0.41	0.39	0.39	0.39			
SUM	14.00	7.33	2.53	2.53				SUM	1	1	1	1	1			
PCM: Social security (SS)								NPCM: SS/N = 4, C.I. = 0.02, R.I. = 0.90, C.R. = 0.01 < 0.1								
CODE	VR37	VR38	VR39	VR40	VR41			CODE	VR37	VR38	VR39	VR40	VR41	SUM (EV)		
VR37	1	2	2	2	2			VR37	0.33	0.44	0.33	0.25	0.25	0.32		
VR38	½	1	2	2	2			VR38	0.17	0.22	0.33	0.25	0.25	0.24		
VR39	½	½	1	2	2			VR39	0.17	0.11	0.17	0.25	0.25	0.19		
VR40	½	½	½	1	1			VR40	0.17	0.11	0.08	0.13	0.13	0.12		
VR41	½	½	½	1	1			VR41	0.17	0.11	0.08	0.13	0.13	0.12		
SUM	3	4.5	6	8	8			SUM	1	1	1	1	1	1		
PCM: Health promotion (HP)								NPCM: HP/N = 5, C.I. = 0.04, R.I. = 1.12, C.R. = 0.03 < 0.1								
CODE	VR42	VR43	VR44	VR45				CODE	VR42	VR43	VR44	VR45	SUM (EV)			
VR42	1	3	1	1/3				VR42	0.19	0.27	0.18	0.18	0.21			
VR43	1/3	1	½	1/5				VR43	0.06	0.09	0.09	0.11	0.09			
VR44	1	2	1	1/3				VR44	0.19	0.18	0.18	0.18	0.18			
VR45	3	5	3	1				VR45	0.56	0.45	0.55	0.54	0.52			
SUM	5.33	11	5.50	1.87				SUM = 1	1	1	1	1	1			
PCM Economic dimension (ECDm)								NPCM: ECDm/N = 4, C.I. = 0.01, R.I. = 0.09, C.R. = 0.01 < 0.1								

**Table 8 ijerph-20-00259-t008:** Evaluation IDC data of Khon Kaen province in Thailand.

CODE	OD	Unit	PI	NI	Khon Kaen	CODE	OD	Unit	PI	NI	Khon Kaen
VR01	MAX	Year	85.29	73.2	75.20	VR23	MAX	%	60	-	25
VR02	MIN	%	5	20	6.28	VR24	MIN	Index	1	100	77.71
VR03	MIN	1K	-	7.6	7.12	VR25	MAX	%	40	-	11.23
VR04	MIN	1K	12	25	1.92	VR26	MAX	Index	5	2	3.03
VR05	MIN	100K	5.6	23.5	13.09	VR27	MIN	%	1	4	3.55
VR06	MIN	100K	8.5	11	4.29	VR28	MAX	%	92	-	79.11
VR07	MIN	100K	38	319	9.69	VR29	MAX	%	98	-	100
VR08	MIN	100K	5.2	34.31	61.72	VR30	MAX	%	100	-	100
VR09	MIN	100K	0.09	20.95	2.67	VR31	MAX	%	90	-	95.08
VR10	MIN	100K	71.26	131.53	127.39	VR32	MAX	%	100	-	1
VR11	MIN	100K	21.77	308.08	50.91	VR33	MIN	100K	14.02	95.61	17.10
VR12	MIN	100K	-	112.37	24.01	VR34	MIN	100K	0.16	2.01	39.90
VR13	MIN	100K	-	18.5	30.52	VR35	MIN	100K	2.64	14.99	28.85
VR14	MIN	100K	-	46.3	4.90	VR36	MIN	%	0.4	5.4	0.56
VR15	MIN	100K	-	77.01	20.05	VR37	MAX	%	80	-	79.96
VR16	MAX	100K	170	10	84	VR38	MAX	%	80	-	82.87
VR17	MAX	100K	300	110	280	VR39	MIN	%	1.82	8.52	2.27
VR18	MAX	100K	380	60	310	VR40	MAX	%	80	-	17.97
VR19	MAX	100K	6	1.7	1.50	VR41	MAX	%	80	-	46.37
VR20	MAX	100K	780	210	381.80	VR42	MIN	%	-	6.5	2.15
VR21	MAX	100K	3.3	1	8.80	VR43	MIN	%	-	8.44	6.63
VR22	MAX	%	100	90	99.62	VR44	MIN	%	-	43	44.99
						VR45	MAX	%	3.3	-	2.04

Note: 1K = 1000 Population.

**Table 9 ijerph-20-00259-t009:** *A*+ and *A*− for each IDC.

CODE	*A+*	*A−*	CODE	*A+*	*A−*	CODE	*A+*	*A−*
VR01	0.30	0.26	VR16	0.29	0.00	VR31	0.00	0.09
VR02	0.03	0.10	VR17	0.08	0.06	VR32	0.00	0.10
VR03	0.02	0.15	VR18	0.10	0.01	VR33	0.01	0.07
VR04	0.00	0.09	VR19	0.27	0.04	VR34	0.00	0.15
VR05	0.02	0.08	VR20	0.05	0.00	VR35	0.03	0.34
VR06	0.04	0.49	VR21	0.04	0.00	VR36	0.03	0.39
VR07	0.02	0.19	VR22	0.06	0.07	VR37	0.23	0.00
VR08	0.01	0.10	VR23	0.07	0.00	VR38	0.17	0.00
VR09	0.01	0.07	VR24	0.00	0.17	VR39	0.04	0.18
VR10	0.08	0.24	VR25	0.46	0.00	VR40	0.12	0.00
VR11	0.00	0.06	VR26	0.19	0.07	VR41	0.10	0.00
VR12	0.00	0.15	VR27	0.08	0.30	VR42	0.00	0.20
VR13	0.01	0.28	VR28	0.20	0.00	VR43	0.00	0.07
VR14	0.01	0.05	VR29	0.04	0.00	VR44	0.00	0.13
VR15	0.00	0.05	VR30	0.00	0.04	VR45	0.44	0.00

**Table 10 ijerph-20-00259-t010:** Deviation between HIS of each IDC.

CODE	*S^+^*	*S^−^*	CODE	*S^+^*	*S^−^*	CODE	*S^+^*	*S^−^*
VR01	0.001	0.000	VR16	0.063	0.001	VR31	0.037	0.001
VR02	0.000	0.005	VR17	0.001	0.000	VR32	0.001	0.029
VR03	0.015	0.000	VR18	0.005	0.001	VR33	0.000	0.002
VR04	0.000	0.007	VR19	0.040	0.000	VR34	0.002	0.000
VR05	0.001	0.001	VR20	0.002	0.000	VR35	0.008	0.000
VR06	0.022	0.090	VR21	0.000	0.001	VR36	0.000	0.010
VR07	0.029	0.000	VR22	0.000	0.000	VR37	0.000	0.003
VR08	0.003	0.002	VR23	0.002	0.001	VR38	0.022	0.000
VR09	0.001	0.001	VR24	0.016	0.001	VR39	0.098	0.000
VR10	0.016	0.001	VR25	0.110	0.017	VR40	0.000	0.120
VR11	0.000	0.001	VR26	0.005	0.001	VR41	0.000	0.051
VR12	0.001	0.014	VR27	0.037	0.001	VR42	0.004	0.018
VR13	0.074	0.000	VR28	0.001	0.029	VR43	0.003	0.000
VR14	0.001	0.001	VR29	0.000	0.002	VR44	0.017	0.000
VR15	0.002	0.000	VR30	0.002	0.000	VR45	0.029	0.075

**Table 11 ijerph-20-00259-t011:** The relative closeness and ranking of each factor and dimension.

Dimension	Element	CODE	*S^+^*	*S^−^*	$c_{}^{*}$	Rank
	HS	VR01–VR05	0.13	0.11	0.47	2
	CDC	VR06–VR09	0.23	0.30	0.57	1
	NCDs	VR10–VR15	0.31	0.13	0.30	3
	HR	VR16–VR22	0.33	0.06	0.15	4
HEDm	-	VR01–VR22	0.52	0.36	0.40	(3)
	ERM	VR23	0.04	0.03	0.42	1
	APM	VR24	0.13	0.04	0.23	4
	PNA	VR25	0.33	0.13	0.28	3
	WM	VR26	0.07	0.04	0.34	2
ENDm	-	VR23–VR26	0.37	0.14	0.28	(4)
	HSS	VR27–VR32	0.22	0.20	0.48	3
	SS	VR33–VR36	0.35	0.35	0.50	2
	HP	VR37–VR41	0.10	0.32	0.76	1
SODm	-	VR27–VR41	0.42	0.52	0.55	(2)
	CEM	VR42	0.07	0.13	0.67	1
	PR	VR43	0.06	0.02	0.21	3
	HHD	VR44	0.13	0.00	0.00	4
	ECG	VR45	0.17	0.27	0.62	2
ECDm	-	VR42–VR45	0.23	0.30	0.57	(1)

## Data Availability

Not applicable.

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
