# Peer review of "The Dynamic Evaluation Model of Health Sustainability under MCDM Benchmarking Health Indicator Standards"

_ijerph, 2022, doi:10.3390/ijerph20010259_

Round 1

Reviewer 1 Report

Manuscript title: The Dynamic Evaluation Model of Health Sustainability Under MCDM Benchmarking Health Indicator Standard.

In this study, the authors present a hybrid sustainability assessment model that integrates the health dimension with the three dimensions generally used in the study on sustainable development and develops an assessment method to evaluate real data with references to the Health Indicator Standard from reliable organizations. For this need, the authors used multi-criteria decision-making methods: AHP-TOPSIS. The essence of this paper is in the case studies. I did not recognize methodological contributions in the paper. The manuscript is well organized, and the contents fit with the journal’s topics.

However, it also presents several flaws that need to be addressed before considering it for publication:

1) Describe the contributions to your paper in more detail in the introduction section.

2) In the last paragraph of the Introduction, the structure of the article should be discussed.

3) The letters and numbers in table 1 are poorly visible, they are too small.

4) The flowchart (Figure 1) of the methodology has many details, which is why the data is less visible. I suggest the authors improve figure 1 a bit.

5) How did you get the data in table 7? Did you use experts? If so, who are they, how many were there, what are their competencies, and how did you aggregate their opinions? From table 7, it doesn't see that you used experts.

6) Sensitivity analysis is missing. I think that for this case study, an analysis by changing the weight coefficients of the criteria would be very useful. This should be shown, compared, and discussed. See more about sensitivity analysis in the paper: Gorcun, O. F., Senthil, S., & Küçükönder, H. (2021). Evaluation of tanker vehicle selection using a novel hybrid fuzzy MCDM technique. Decision Making: Applications in Management and Engineering, 4(2), 140-162. https://doi.org/10.31181/dmame210402140g;

7) Can the results be compared with some earlier research?

8) In the Conclusion section, show in detail the advantages and limitations of the proposed methodology and this study.

9) The authors need to provide several solid future research directions clearly.

10) Some references are incomplete (for example, 26 and 27)

Author Response

Dear reviewer 1,

               First, thank you for your precious comments and suggestions for our research. we have tried to improve according to your suggestions and provided answers to your questions as attached file.

Kind regards 

Nutthawut Ritmak 

Reviewer 2 Report

The topic is timely and interesting. The authors have paid good attention to details. But I have concerns about the methodological aspects of this paper. I strongly recommend that the authors address the following.

1- There is no information from experts. What is their profile (education, work experience, the field of expertise, etc.)? A method based on expert preferences has been used to achieve the results, and these are very important.

2- The results indicate that the opinions of an expert were used. Given the huge number of indicators, how are the preferences of only one expert taken into account? Why the researchers didn't use the group decision-making process (GAHP, etc.)?

3-  The hierarchical structure of the investigated problem includes three levels. Dimensions, elements, and indicators. The indicators are weighted by the AHP method.

3-1 The importance of dimensions and elements has not been considered. Is their importance the same? In this situation, the local weight of the indicators can be calculated and their global weight cannot be calculated.

3-2 To obtain the global weight of the indicators, the weight of the indicators must be multiplied by the weight of the upper level (elements).

3-3 In Figure 4, in the output part of the research, result 3 and result 4 are shown in continuous space. Only the local weights of the indicators have been calculated and the global weights (the importance of each indicator among all 45 indicators, not only the importance of each indicator in the relevant element) have not been calculated. In this case, only the importance of each index in the corresponding element can be explained because the importance of the dimensions is considered equally.

4- In section 3. The data are somewhat that's not satisfactory to me. to create a decision matrix for the TOPSIS method, Where did the data (Xmn) come from?? If the data is quantitative, show it as a supplementary file. If the priority of alternatives has been qualitatively measured for each indicator, what scheme was used, and who was the decision-maker?

Author Response

Dear reviewer 2,

               First, thank you for your precious comments and suggestions for our research. we have tried to improve according to your suggestions and provided answers to your questions as attached file.

Kind regards 

Nutthawut Ritmak 

Reviewer 3 Report

The authors have dedicated this paper to a very interesting subject and discuss the possibility to create a dynamic evaluation Model of Health Sustainability. Since the evaluation discussed Thailand - this should be indicated in the title. 

Reviewer has found some data lacking in the methodology of presented research. The authors write that: "To select criteria (CTR), as hereafter called indicators (IDC), the researchers firstly gather variables from literature reviews and related standards such as ISO37120, ISO37122, ISO37123, U4SSC. [27] Considering the context of Thailand, the researchers also compile quantitative secondary data from the government to improve the criteria."  Ref. 27 - is a single position in References, and in order to gather variables much more data, including ISO standards was probably reviewed. These should also be added. If the researchers compile date from the government - then they also give adequate reference as to where from and what was collected to the purpose of this paper. Also please explain in detail as to what does "improvement of criteria imply". If  Figure 1 supports the line of research, then it should be placed nearer methodology section and explained in detail.

The choice of Khon Kaen should be better supported, other criteria - except for the brief mention that the city has implemented a strategy aiming to be  a healthcare center in 2023.

Discussion section is missing. There is no such section as "Suggestions" - please refer to Author's support data. Also please explain the limitation of presented analysis.

Please check the references - in some cases they are not presented according with IJERPH expectations.

Author Response

Dear reviewer 3,

               First, thank you for your precious comments and suggestions for our research. we have tried to improve according to your suggestions and provided answers to your questions as attached file.

Kind regards 

Nutthawut Ritmak 

Round 2

Reviewer 1 Report

All the reviewers' comments have been addressed carefully and sufficiently. The revisions are rational from my point of view. I think the current version of the paper can be accepted.

Author Response

Dear reviewer, 

Thank you for providing comments for our paper. The comments have helped deepened and broaden our curiosity in the field with an inspiration to conduct insightful work in the future. Wish you all the best and Merry Christmas!

Kind regards,

Nutthawut Ritmak 

Reviewer 2 Report

An important issue remains unanswered. The authors are expected to address this issue. To explain this issue more clearly, I will state it as follows:

The group of experts including fifteen experts contributed in two parts of the article. The first part, To examine 45 indicators used in the research and confirm the suitability of the indicators for the decision making process. There is no ambiguity at this point. In the second part, the experts make pairwise comparisons between the indicators in order to obtain the final weights of the indicators and express their preferences using the nine-point scale of saaty.

- In Table 7, where the final weights of the indicators are analyzed, which of the 15 experts expressed their opinions? Because the preferences of only one expert are shown in Table 7(in the form of 1-9 scale).

- If Table 7 is filled by an expert, the results are not acceptable due to the number of indicators and the complexity of the decision-making problem.

- If more than one expert participated in pairwise comparisons, there will be a "Table 7" for each expert. More clearly, for each indicator, 15 different weights will be obtained for each expert. And in this case, the final weights of the indicators can be determined with a simple average of the final weights obtained for each expert.

Author Response

Dear reviewer, 

    Thank you for providing comments for our paper. The comments have helped deepened and broaden our curiosity in the field with an inspiration to conduct insightful work in the future. Wish you all the best and Merry Christmas!. We have tried to provided answers for your questions in the attached file.

Kind Regard

Reviewer 3 Report

I believe that the responses given to the reviewers were shallow and did not involve any additional value. Hence I evaluate this paper as falling within the average level of interest. The authors made minor level of changes to their paper - but since they have actually done something - I accept the paper the way I has been presented.

Author Response

(The authors gave the same response as above.)
